

# Balanced and positively worded personality short-forms: Mini-IPIP validity and cross-cultural invariance

Agustín Martínez-Molina[1] and Víctor B. Arias[2]

[1] Departamento de Psicología y Sociología, Universidad de Zaragoza, Teruel, Spain
[2] Facultad de Psicología, Universidad de Salamanca, Spain

## ABSTRACT

**Background**. The Mini-IPIP scales (*Donellan et al., 2006*) are possibly one of the most commonly used short inventories for measuring the Big Five Factors of personality. In this study, we aimed to investigate the psychometric properties of two Mini-IPIP Spanish short forms, one balanced and one positively wording (PW).

**Method**. Two samples, one from native Spanish speakers and another from native English speakers, made up a total of 940 participants in this study. The short forms were translated and adapted based on international guidelines. Reliability (internal and composite) and validity analyses (construct ESEM, concurrent, predictive and cross-cultural invariance through multi-group factorial models) were performed.

**Results**. For both the balanced scale and the PW one, modeling a method factor was not relevant. The reliability and validity indices of both forms were according to theory and prior studies' findings: (a) personality factors were medium-high related to affective factors; (b) personality factors were less related to life satisfaction than affective factors; (c) life satisfaction was medium-high related to affective factors; (d) neuroticism appeared mainly related to all criteria variables; and (e) an acceptable level of invariance was achieved with regard to the English version.

**Discussion**. This study contributes to research on personality assessment by providing the first evidence regarding the psychometric properties of a PW short measure. These results suggest that PW short scales of personality used after data screening techniques may be appropriate for future studies (e.g., cross-cultural, content validity).

## INTRODUCTION

Long testing sessions can lead to exhaustion, irritation, and demotivation in the respondent, particularly in low-stakes contexts (*Wise & DeMars, 2005*). Such conditions increase the probability of diminished attention to the task and produce responses that are not based on the item content (e.g., careless responding; *Meade & Craig, 2012*), with a consequent deterioration of the validity and a need to implement control measures that further prolong the testing session (*DeSimone, Harms & DeSimone, 2015*; *Maniaci & Rogge, 2014*; *Oppenheimer, Meyvis & Davidenko, 2009*; *Weijters, Baumgartner & Schillewaert, 2013*).

Corresponding author
Agustín Martínez-Molina,
agustin@unizar.es

As a consequence, possessing short-form measures that can deliver an acceptable proxy for a person's position on broad constructs is clearly of interest in terms of both research and applied assessment (*Credé et al., 2012*). This is particularly true in the case of multidimensional and multifaceted constructs such as human personality, the assessment of which has traditionally required extensive scales. In recent years, substantial efforts have been made to develop more efficient measures of basic personality traits. A representative example can be found in the diverse levels of reduction of scales evaluating the Big Five based on the International Personality Item Pool (IPIP; *Goldberg, 1999*): 300 items (IPIP-NEO; *Goldberg, 1999*), 120 items (IPIP NEO 120; *Johnson, 2014*), 50 items (IPIP Big Five Factor markers; *Goldberg, 1992*), 32 items (IPIP-IPC; *Markey & Markey, 2009*), and 20 items (Mini-IPIP; *Donellan et al., 2006*).

The Mini-IPIP is possibly one of the most commonly used short scales based on the Big Five model. *Donellan et al. (2006)* developed the Mini-IPIP with five main objectives: (a) to achieve adequate balance between brevity and sound psychometric properties, (b) to ensure a sufficient number of items per factor in order to permit their non-problematic application in structural equation modelling, (c) to maximize the empirical independence between the five basic factors (i.e., correlations between factors close to zero), (d) to maximize the discriminant validity of the items (i.e., that they would be nearly pure indicators of their theoretical factor), and (e) to minimize losses in reliability and validity with regard to longer versions of the scale. The results of the five studies described by *Donellan et al. (2006)* suggest that the objectives above were met adequately. The Mini-IPIP showed to be a reasonable proxy for evaluating the Big Five factors and for offering an adequate trade-off between conciseness, reliability, and both construct and criterion validity. However, the authors did not obtain adequate fit in the confirmatory factor models, a problem that is common in complex multidimensional models of personality (*Hopwood & Donnellan, 2010*), possibly due to the excessive restrictiveness of the independent cluster model of confirmatory factor analysis (*Marsh et al., 2010*).

Although the Mini-IPIP has been widely used in research since 2006, few studies have been explicitly dedicated to evaluating its psychometric properties (*Oliveira, 2017*). This is important considering the essential role of an independent replication of results for guaranteeing the validity of short-forms (*Smith, McCarthy & Anderson, 2000*). *Cooper, Smillie & Corr (2010)* found poor fit with a five correlated factors confirmatory model, as well as a model with two superordinate factors. However, they obtained acceptable internal consistency coefficients (Cronbach's alpha), and exploratory factor analysis (EFA) supported the empirical independence of the dimensions (i.e., low correlations) and the discriminant validity of the items (i.e., the primary loadings were substantially higher than the cross-loadings). *Baldasaro, Shanahan & Bauer (2013)* found similar results with a sub-optimal fit in the CFA models, with several high modification indices suggesting possible violations of local independence between pairs of items. They also tested measurement invariance (sex and ethnicity) without exceeding metric invariance in any of the sub-scales. Cross-cultural differences in self-reported items were usually analyzed through expanding an invariance test to recognize possible cultural variance in language; that is, to analyze if the answers to the same item, although in different languages, is understood in an equal or

similar manner. *Laverdière, Morin & St-Hilaire (2013)* found poor fit with the confirmatory model, requiring the freeing of correlations between three pairs of residuals because of semantic similarity of items belonging to the same facet. They also found reasonable evidence of measurement invariance to a level of variances/covariances (undergraduates vs. employees, age, and gender).

Personality scales have been balanced traditionally as an intent to cancel out the effects of the agreeing tendency (*Couch & Keniston, 1960*). Half of the items are worded positively and the other half are worded negatively. Balance keying as a method control for acquiescence can improve the psychometric properties of the personality measures (*Konstabel et al., 2017*; *Rammstedt, Kemper & Borg, 2013*). However, method effects associated with worded items may emerge producing stable response-style factors (*Kam & Chan, 2018*; *Marsh, Scalas & Nagengast, 2010*), i.e., systematic artifacts with undesirable consequences for measurement produced by the inclusion of a mix of positively and negatively keyed items.

Wording effects occur when we assume that some direct and inverse items are equivalent in their ability to reflect a construct, and then response patterns appear in the positive or negative wording of the items instead of in their content (*Podsakoff et al., 2003*). These response patterns are not random and can lead both to the artificial inflation of the correlations between latent variables, and to the deflation of the correlation between the items of the same factor (*Huang, Liu & Bowling, 2015*). The result can lead to a multidimensionality scenario that frequently requires extra method factors (*Arias & Arias, 2017*; *Eid, 2000*), or the implementation of various cleaning data procedures (*Meade & Craig, 2012*).

On personality short scales, the wording effect is even less known. Wording effects have been considered as one of the possible variables that causes detriments of the psychometric scales' properties (e.g., overestimated test reliability, misfit validity; *Eys et al., 2007*; *Lai, 1994*; *Wang, Chen & Jin, 2015*). That being said, it should be noted that over half of the original Mini-IPIP items use negative wording (i.e., due to the use of words with inverse semantic polarity to the measured trait or by the use of the adverb "not"). Based on previous research we hypothesized that the use of negative items is not relevant in these brief scales, that is, without validity and reliability consequences.

Although Spanish adaptations of the IPIP Big Five markers do exist (*Cupani, 2009*; *Cupani & Lorenzo-Seva, 2016*; *Goldberg, 1992*) and could be used to obtain translations of the Mini-IPIP items, to our knowledge, no study to date has assessed the psychometric properties of the Mini-IPIP in Spanish nor investigated its measurement equivalency with regard to the original English version. This point is particularly relevant, given that, unless it is demonstrated that the scale is measuring the same construct in the same manner, it is not possible to guarantee that the translated version is truly equivalent to the original measure as designed by its creators (*Wu, Li & Zumbo, 2007*).

The purpose of this study was to investigate the psychometric properties of two translated and adapted Mini-IPIP Spanish short forms, one balanced and one positively worded (PW). To this end, we: (a) adapted the Mini-IPIP scales from the English to Spanish following internationally recognized quality standards (*Muñiz, Elosua & Hambleton, 2013*); (b) proposed a parallel positively-worded version of these scales; (c) verified, in both

versions, that the dimensionality, internal structure, and other evidence of validity and reliability correspond to expectations based on theory and prior studies (i.e., concurrent and predictive indexes with emotions and life satisfaction); and (d) investigated measurement invariance with regard to the English version through multi-group factorial models.

## METHOD

### Participants

Two samples were used. Sample 1 (native Spanish speakers) was collected to assess the validity and reliability of the proposed scales. Sample 2 (native English speakers) was obtained to analyze the cross-cultural invariance.

Sample 1 comprised 560 students enrolled in five different faculties at two Chilean universities. The evaluation was completely anonymous, and consent was obtained from all participants for their responses to be used as part of the research. Sample 1 scales were computer-lab administered in groups of 15 participants where one of the authors of this work was always present.

Sample 2 consisted of 380 native English speakers of US nationality with diverse levels of educational attainment (no formal qualification: 5%; secondary school: 19%; college: 28%; undergraduate degree: 37.5%; graduate degree: 8.5%; doctoral degree: 2%). Sample 2 data were gathered through Prolific Academic, a service supported by Oxford University that specializes in online data gathering using panels of participants defined in advance by the researcher.

### Procedure

Before starting the study, ethical approval was obtained from the Bioethics Committee of Universidad de Talca (projects no. 1151271, no. 11140524). In both applications, (a) the participants received monetary compensation equivalent to 2 USD, (b) the items were presented with the same visual arrangement and order, (c) the acceptance rate was 100% (no missing data were observed), and (d) to identify the participants that responded inattentively, both samples underwent an instructional manipulation check (IMC; *Oppenheimer, Meyvis & Davidenko, 2009*; *Weijters, Baumgartner & Schillewaert, 2013*). The IMC consisted of an item which displayed an identical rating scale as the rest of the items but contained a specific instruction ("For this statement, please do not check a response option"). The participants who, despite the special instruction, responded to the item were considered careless responders and thus were eliminated from the analysis.

A total of 45 students did not meet the criterion of attentional control in this study ($n = 32$ in Sample 1, $n = 13$ in Sample 2). The final sampling sizes were: Sample 1, $n = 518$ (age range $= 18$–$34$, $M = 21.6$, SD $= 2.3$; 70.7% women), and Sample 2, $n = 367$ (age range $= 18$–$72$, $M = 34.6$, SD $= 12.7$; 48.5% women).

For efficiency reasons and validity purposes a fraction of Sample 1 (50%, $n = 280$) was randomly selected to complete three more short-forms scales. One out two assessment groups of participants were asked to complete two extra scales (Sample 1 after attentional control, $n = 278$).

## Measures

*The Mini-IPIP Scales* (*Donellan et al., 2006*): This is the abbreviated version of the IPIP Big Five factor markers (*Goldberg, 1992*), which consists of 20 items (four per dimension): Extraversion (E), Agreeableness (A), Conscientiousness (C), Neuroticism (N), and Openness (O). Eleven items are reverse-scored. Each item is answered by the participant on a five-point accuracy scale (1 = Not at all, 5 = Completely) in accordance with the level to which each statement is applicable to their habitual behavior. As reported in four studies by *Donellan et al. (2006)*, the average reliabilities indices (Cronbach's $\alpha$) of this version are .81 (E), .73 (A), C (.70), N (.74) and O (.69). Correlation and fit indices also supported the construct, convergent and discriminant validity according to other Big Five measures.

*The Mini-IPIP Scales Spanish version*: This form was created on the base of the International Test Commission guidelines for translating and adapting tests (*ITC, 2005*; *Muñiz, Elosua & Hambleton, 2013*). Three native-Spanish speakers with advanced English proficiency and academic expertise in the field (i.e., proficient level C in English according to the Common European Framework of Reference for Languages and doctoral studies in measurement or personality assessment) considered linguistic and cultural factors in the translation and adaptation of the original Mini-IPIP items (*Donellan et al., 2006*). An iterative debug translating procedure was performed. The experts began with independent translations that they iteratively shared until they rationally agreed upon a final debugged version. The translations were highly concordant in their content in the first iteration. Items that maintained the original meaning and contained only standard Spanish expressions were selected. The experts did not reach sufficient agreement on two items in the second iteration (item 18 ''make a mess of things'' and item 19 ''seldom feel blue''). A pilot test on 73 volunteers (convenience sampling) with socio-demographic characteristics of age and sex similar to those of the final participants (sample 1) was conducted to provide empirical evidence to resolve the discrepancies in the mentioned items (18 and 19). For these items, we selected the wording (from two alternatives for each item) that showed the higher factorial loading in their main factor and lower cross-loadings, according to an EFA based on a polychoric correlations matrix with unweighted least squares estimator and promax rotation. Appendix A and B show the final versions of the Mini-IPIP with the basic descriptive statistics obtained from the two main samples of this study and instructions that the participants received.

*The Mini-IPIP Scales positively worded (PW) Spanish version*: Following the same adaptation method described above, a positively worded set of items was composed (see Appendix B). In order to have a complete positive short-form of the scales we made a parallel version from the original reverse 11 items (the rest of items were already PW in the original version). Of these items, those of N were elaborated with a positive semantic polarity towards ''Emotional Stability'' instead of ''Neuroticism''.

*The PANAS-C10 Spanish version*: In accordance with the framework of this study, a short-form was also used to measure positive affect (PA) and negative affect (NA) factors for convergent validity purposes. The original 10-item version (*Damásio et al., 2013*) is a brief form of the PANAS (*Carvalho et al., 2013*; *Watson, Clark & Tellegen, 1988*). Participants were asked to rate two self-report semantic mood scales (five-point ordinal

agreement alternatives; 1 = Not at all, 5 = Completely). The PANAS-C10 English version reported that the reliability indices (Cronbach's $\alpha$) were .81 (PA) and .82 (NA). Fit indices also supported the unidimensionality (62% explained variance, KMO = .82; Barlett (45) = 494.4 $p \leq$ .001; CFI = .97; CFI = .97; RMSR = .06; RMSEA = .04; only one averted dimension with parallel analysis). The PANAS-C10 Spanish version was translated and adapted following the same procedure described for the Mini-IPIP Spanish version. Given the semantic simplicity of this measure (only 10 common words, see Appendix C), it was not surprising that there was almost no disagreement between experts in the item creation process. One item (number 4 "Fun") did not retain its original meaning in its translated version. In Spanish, it is not common to refer "fun" as a personal state of an emotion or feelings. This word is commonly used to refer to circumstances or people who create fun. The synonym "Entretenido" was agreed to as an alternative of "Fun", which is also naturally understood in both versions of the verb "to be" in Spanish (i.e., "ser" and "estar"). PANAS was related to different personality scales for convergent validity support (*Watson & Clark, 1999*). Two Principal Component Analyses grouped these measures accordingly (NA was substantially related with N, and PA to E). Similar results were showed by *Bruck & Allen (2003)*; only N (.69), E (−.16), and A (−.27) reported significant relations with NA.

*The SWLS Spanish version* (*Moyano-Díaz, Martínez-Molina & Ponce, 2014*): The Satisfaction with Life Scale was translated and adapted from *Diener et al. (1985)*. Five items with a response scale of five levels of satisfaction (1 = Not at all, 5 = Completely). The reliability index (Cronbach's $\alpha$) of this version ranged for .82 to .87. Fit indices also supported the unidimensionality (64% explained variance, KMO = .84; Barlett (10) = 576.0 $p \leq$ .001; GFI = 1.0. and RMSR = .035; only one averted dimension with parallel analysis). This short-form was used in this study for predictive validity purposes. *Hayes & Joseph (2003)*, provided evidence that C, N and, E, were related to this subjective measure of well-being (only N and C predicted significantly). Other authors (*Chico, 2006*; *Joshanloo & Afshari, 2011*), found that N and E were related strongly to the SWLS (N accounted most of the variance in both studies).

## Data analysis
### Measurement models
First, the data from Sample 1 were fitted to the Big Five model (*Costa Jr & McCrae, 1992*) through exploratory structural equation modelling (ESEM; *Asparouhov & Muthén, 2009*) with oblique target rotation. The ESEM is a general technique for factorial analysis that permits the estimation of all possible cross-loadings. ESEM offers more precise estimators of the factor loadings and correlations between factors than CFA. ESEM has been shown to be more efficient than CFA in the estimation of complex models with interstitial relationships between items belonging to different dimensions, and furthermore offers the main advantages of confirmatory analysis while maintaining the flexibility of unrestricted factor analysis (*Garrido et al., 2018*; *González-Arias et al., 2018*; *Morin, Arens & Marsh, 2016*; *Marsh et al., 2014*). Target rotation was used in all the models, permitting the specification of a matrix of primary loadings, enabling the use of ESEM in a confirmatory manner (*Asparouhov & Muthén, 2009*).

Four models were estimated for each version of the scales (i.e., Mini-IPIP Spanish and Mini-IPIP Spanish PW). Model 1 (M1) and M1-PW replicated the basic theoretical structure of the scale through the specification of five correlated factors.

M2 and M2-PW specified a series of correlated residuals or Correlated Uniqueness (CU), in order to prevent the estimation of the substantive loadings from being biased by the presence of spurious variance due to semantic similarity between certain pairs of items (*Cole, Ciesla & Steiger, 2007*). CU were freed between residuals of items that simultaneously (a) pertained to the same facet, (b) demonstrated strong similarity of wording, and (c) demonstrated extreme modification indices in M1. In M2 two pairs of items met the three conditions ("Am not interested in abstract ideas" / "Have difficulty understanding abstract ideas," and "Sympathize with others' feelings" / "Feel others' emotions"). M2-PW contained the same CU specifications as M2.

In M3 and M3-PW, a random intercept confirmatory factor was also tested (as a method factor). Random intercept confirmatory factor analysis (RI-IFA; *Billiet & McClendon, 2000*; *Maydeu-Olivares & Coffman, 2006*; *Aichholzer, 2014*) consists of the inclusion of a factor common to all items. The RI factor is orthogonal to the substantive dimensions, and its loadings are fixed to equality (as a consequence, the RI factor occupies a single degree of freedom, corresponding to its variance). The RI factor imposes an artificial relationship between the items with different wording polarity, capturing and isolating the variance associated with response artefacts such as acquiescence (*Maydeu-Olivares & Coffman, 2006*).

M4 and M4-PW added CU and RI specifications of previous models. Finally, Model 5 (M5), based on the English-speaking sample, was structurally identical to M4.

### Cross-cultural invariance analysis

Second, for the selected model in the previous phase, a series of multi-group ESEM models nested in successive levels of increasing restriction were estimated (*Meredith, 1993*; *Millsap & Yun-Tein, 2004*). The tested models were, in the following order, Configural, Strong, and Strict. Each of these models tests a series of hypotheses regarding the equivalence of the parameters between the groups compared. The Configural test assumes that both groups (Chilean and US samples) have the same number of dimensions and the same configuration of factorial loadings; that is, in both groups, a qualitatively similar construct is being measured.

Strong invariance requires that the factorial loadings as well as the intercepts of the items (the thresholds, in this case) be equal between groups. Achieving strong invariance implies that group differences in the latent scores are due to differences in the trait (not dependent on the scale), and therefore, the scores are not biased against any of the groups. As a result, if strong invariance is achieved, the changes observed in the latent measures of the factor can be interpreted as a function of the change in the latent construct (*Marsh et al., 2010*).

Strict invariance entails that the residual variances of the items must be equivalent between groups. Failure to achieve the strict level can be interpreted as differences in the reliability of the raw scores. As a consequence, achieving this level is not as relevant as fulfilling the rest of the invariance conditions, at least when one is working with latent variables where the measurement error is controlled (*Byrne, 2012*). However, the
above is only true when the residual variances meet the requirement of conditional independence; that is, they must be authentically random (*Wu, Li & Zumbo, 2007*; *Deshon, 2004*; *Vandenberg & Lance, 2000*). As a result, given the logical possibility that part of the error would be systematic in nature (which is plausible given that the Mini-IPIP, like other personality scales, contains sub-groups of items pertaining to the same facet), the Strict test acquires importance (a) to guarantee that the results of the scalar invariance are not biased and (b) to guarantee the unbiased comparability of the raw scores and factor scores.

The analyses described above yielded the estimation of six multi-group models (M6 to M8-PW). The PW-Models were tested using the positively worded items in the Spanish sample *vs.* the balanced English version in the English sample.

All of the models used the Weighted Least Squares and adjusted Mean and Variance (WLSMV) estimator, given the ordered-categorical nature of the raw data (*Beauducel & Herzberg, 2006*). Goodness of fit was evaluated using the comparative fit index (CFI), the Tucker-Lewis Index (TLI), and the root mean square error of approximation (RMSEA). CFI and TLI values $\geq$ .95 are considered adequate, as are RMSEA values <.05 (*Schreiber, 2017*). To compare the fit of nested models in the multi-group analysis, the recommendations of *Chen (2007)* and *Cheung & Rensvold (2002)* were followed ($\Delta$CFI $\geq$ −.01 supplemented by a $\Delta$RMSEA $\geq$ .015 suggest noninvariance). The analyses were performed using MPlus v. 7.4 (*Muthén & Muthén, 2014*) and Winsteps 4.1.0 (*Linacre, 2018*).

# RESULTS

## Estimated models

The fit indices for all the models are shown in Table 1. The ESEM analysis produced a sub-optimal fit in the case of M1 and M1-PW (e.g., RMSEA > .08, TLI < .90). As expected, the modification indices suggested freeing the correlations between the residuals of the commented items. M2 and M2-PW (CU allowed) showed an acceptable fit ($\chi 2$ (98) = 240 and 171, RMSEA = .05; CFI > .97; TLI > .94).

M3 and M3-PW (response artefacts controlled) presented no fit improvement over M2 and M2-PW ($\chi 2$ (99) = 337 and 178, RMSEA < .07; CFI > .95; TLI > .91), suggesting that the wording effect, while present to a certain extent, was concentrated in the commented items and was not a relevant problem in this sample.

M4, M4-PW and M5 (US sample) that added CU and RI specifications achieved adequate fit levels ($\chi 2$ (97) = 235, 172 and 241, RMSEA < .06; CFI > .97; TLI > .95). The estimated factor loading for the RI method factor in M4 and M4-PW were .141 and .085. One of the advantages of modeling the method factor with RI is that the loss of degrees of freedom is minimal (i.e., 1 degree). When the method factor is not relevant, as it was in this case, the fit indices of these models were almost identical.

Among the models tested, the most parsimonious and the ones with the best fit were the M2 and M2-PW. The parameters obtained from this model are shown in Table 2. All the primary loadings were significant (M = .69; DT = 0.13) and their standard errors reasonably reduced (M = .03, SD = 0.01). All the cross-loadings were small (absolute mean = .0; SD = 0.005), and mainly non-significant ($p > .01$). Finally, the reliability of

**Table 1  Fit indices of the estimated models.**

| | Model | Type | RMSEA | CFI | TLI | $\chi 2$ | $df$ | $\Delta$RMSEA | $\Delta$CFI | $\Delta$TLI | $\Delta\chi 2$ | $\Delta df$ |
|---|---|---|---|---|---|---|---|---|---|---|---|---|
| Measurement | M1 | ESEM | .084 | .929 | .864 | 469 | 100 | | | | | |
| | M1-PW | ESEM | .082 | .96 | .924 | 288 | 100 | | | | | |
| | **M2** | **ESEM+CU** | **.053** | **.973** | **.947** | **240** | **98** | | | | | |
| | **M2-PW** | **ESEM+CU** | **.052** | **.984** | **.970** | **171** | **98** | | | | | |
| | M3 | ESEM+RI | .068 | .954 | .912 | 337 | 99 | | | | | |
| | M3-PW | ESEM+RI | .054 | .983 | .968 | 178 | 99 | | | | | |
| | M4 | ESEM+CU+RI | .052 | .973 | .948 | 235 | 97 | | | | | |
| | M4-PW | ESEM+CU+RI | .053 | .984 | .969 | 172 | 97 | | | | | |
| | M5 | ESEM +CU +RI (us) | .064 | .974 | .949 | 241 | 97 | | | | | |
| Invariance | *Language, Spanish-English* | | | | | | | | | | | |
| | M6 | Configural | .057 | .974 | .949 | 478 | 196 | | | | | |
| | **M7** | **Strong** | **.068** | **.938** | **.927** | **994** | **326** | **.011** | **−.036** | **−.022** | **516** | **130** |
| | **M8** | **Strict** | **.069** | **.932** | **.925** | **1079** | **346** | **.012** | **−.042** | **−.024** | **601** | **150** |
| | **M7p** | **Strong (partial)** | **.060** | **.955** | **.944** | **796** | **310** | **.003** | **−.019** | **−.005** | **318** | **114** |
| | **M8p** | **Strict (partial)** | **.060** | **.952** | **.944** | **837** | **326** | **.003** | **−.022** | **−.005** | **359** | **130** |
| | *M6-PW* | *Configural* | *.058* | *.979* | *.960* | *409* | *196* | | | | | |
| | *M7-PW* | *Strong* | *.088* | *.920* | *.907* | *1,141* | *326* | *.030* | *−.059* | *−.053* | *732* | *130* |
| | *M8-PW* | *Strict* | *.091* | *.911* | *.902* | *1,264* | *346* | *.033* | *−.068* | *−.058* | *855* | *150* |
| | ***M7-PWp*** | ***Strong (partial)*** | ***.060*** | ***.967*** | ***.956*** | ***621*** | ***286*** | ***.002*** | ***−.012*** | ***−.004*** | ***212*** | ***90*** |
| | ***M8-PWp*** | ***Strict (partial)*** | ***.070*** | ***.955*** | ***.942*** | ***757*** | ***294*** | ***.012*** | ***−.024*** | ***−.018*** | ***348*** | ***98*** |

**Notes.**

RMSEA, Root Mean Square Error of Approximation; CFI, Comparative Fit Index; TLI, Tucker-Lewis Index; ESEM, Exploratory Structural Equation Model; CU, Correlated Uniqueness; RI, Random Intercept factor; PW, Positive Wording; us, United States sample.

Bold measurement models were selected for the invariance tests; Invariance models by $\Delta$RMSEA $\leq$ .015 are in bold; In italic the invariance results of the PW-Models using the positively worded items in the Spanish sample vs. the regular English version (balanced) in the English sample.

each substantive factor was calculated using the coefficient of composite reliability (*Raykov, 1997*). The five factors acquired good composite reliability (range = .90 to .94). As expected based on the theoretical model (*Costa Jr & McCrae, 1992*) and previous research (*Donellan et al., 2006*; *Baldasaro, Shanahan & Bauer, 2013*), the correlations between factors were low (ranged from –.01 to .27).

## Cross-cultural invariance analysis
### Test of configural invariance
The fit indices for the invariance models are shown in Table 1. Configural structure M6 and M6-PW, based on M2 and M2-PW (ESEM + CU), were the base models (i.e., with which the rest of the invariance models were compared). With the exception of the RMSEA that slightly exceeds the recommended value, M6 and M6-PW produced an acceptable fit (RMSEA < .06; CFI ≥ .95; TLI ≥ .95).

### Test of strong invariance
M7 strong invariance test was executed imposing equality restrictions on the 80 thresholds corresponding to the five categories of each item (i.e., scalar invariance). Between samples (Spanish-English) the balance version of the scales showed that the change in the RMSEA satisfied the criterion of invariance while the CFI did not (M7: $\Delta\chi 2 = 516$; $\Delta df = 130$;

**Table 2  Factor loadings, correlations and reliability of M2 and M2-PW (ESEM + CU).**

| Factor | # | F1 | | F2 | | F3 | | F4 | | F5 | |
|---|---|---|---|---|---|---|---|---|---|---|---|
| | | R | PW | R | PW | R | PW | R | PW | R | PW |
| E | 1 | **.824** | **.802** | – | – | – | – | – | – | – | – |
| | 6 | **.730** | **.756** | – | – | – | – | – | – | – | – |
| | 11 | **.784** | **.794** | – | – | – | – | – | – | – | – |
| | 16 | **.718** | **.739** | – | – | – | – | – | – | – | – |
| N | 4 | – | – | **.732** | **.740** | – | – | – | – | – | – |
| | 9 | – | – | **.568** | **.466** | – | – | .258 | – | – | – |
| | 14 | – | – | **.634** | **.568** | – | – | – | – | – | – |
| | 19 | – | – | **.543** | **.667** | – | – | – | – | – | – |
| A | 2 | – | – | – | – | **.598** | **.791** | – | – | – | – |
| | 7 | – | – | – | – | **.800** | **.913** | – | – | – | – |
| | 12 | – | – | – | – | **.500** | **.836** | – | – | – | – |
| | 17 | – | – | – | – | **.806** | **.845** | – | – | – | – |
| C | 3 | – | – | – | – | – | – | **.557** | **.593** | – | – |
| | 8 | – | – | – | – | – | – | **.688** | **.776** | – | – |
| | 13 | – | – | – | – | – | – | **.711** | **.724** | – | – |
| | 18 | – | – | – | – | – | – | **.946** | **.945** | – | – |
| O | 5 | – | – | – | – | – | – | – | – | **.706** | **.831** |
| | 10 | – | – | – | – | – | – | – | – | **.480** | **.498** |
| | 15 | – | – | – | – | – | – | – | – | **.567** | **.477** |
| | 20 | – | – | – | – | – | – | – | – | **.878** | **.959** |
| | CR | .94 | .93 | .90 | .87 | .92 | .94 | .93 | .92 | .91 | .90 |
| | F1 | – | – | | | | | | | | |
| | F2 | −**.270** | −.146 | – | – | | | | | | |
| | F3 | **.251** | .059 | −**.145** | .060 | – | – | | | | |
| | F4 | −.010 | .009 | −.101 | −.043 | **.164** | .102 | – | – | | |
| | F5 | **.234** | **.205** | −.107 | −.033 | **.193** | .131 | −.022 | −.069 | – | – |

Notes.

E, Extraversion; A, Agreeableness; C, Conscientiousness; N, Neuroticism; O, Openness; #, Item administration order; R, Regular version; PW, Positively worded version; CR, Composite reliability.

Loadings >.20 and $p < 0.01$ are shown; Main loadings are in bold; Factor correlations $p < 0.01$ are in bold.

$\Delta$RMSEA = .011; $\Delta$CFI = −.036; $\Delta$TLI = −.022). The invariance analysis showed local misfit (thresholds with standardized expected parameter changes >.20 and modification indices >10; *Garrido et al., 2018*; *Saris, Satorra & Van der Veld, 2009*; *Whittaker, 2012*). A partial invariance model (M7p) was executed following a stepwise implementation. One at a time, the threshold with the highest outlier value was constricted. Even though the CFI-threshold criterion was not reached in M7p, the $\Delta$RMSEA did (M7p: $\Delta\chi2$ = 318; $\Delta df$ = 114; $\Delta$RMSEA = .003; $\Delta$CFI = −.019; $\Delta$TLI = −.005).

As we pointed out before, the PW-Models were tested using the positively worded items in the Spanish sample *vs.* the balanced English version in the English sample. M7-PW strong invariance test was modeled by imposing the same initial constraints on the thresholds as M7. In this invariance test the change of the two fit indices did not meet the

recommended criteria (M7-PW: $\Delta\chi2 = 732$; $\Delta df = 130$; $\Delta$RMSEA $= .030$; $\Delta$CFI $= -.059$; $\Delta$TLI $= -.053$). The M7-PWp was also tested following the same constriction process as M7p. The change in RMSEA meet the invariance threshold and the change in CFI was very close to achieving it (M7-PWp: $\Delta\chi2 = 212$; $\Delta df = 90$; $\Delta$RMSEA $= .002$; $\Delta$CFI $= -.012$; $\Delta$TLI $= -.004$).

### Test of strict invariance

From the strong invariance model, the residual variances were fixed to equality. As shown in Table 1, the RMSEA satisfied the criterion of invariance while the CFI did not (M8: $\Delta\chi2 = 601$; $\Delta df = 150$; $\Delta$RMSEA $= .012$; $\Delta$CFI $= -.042$; $\Delta$TLI $= -.024$). Like M7-PW, M8-PW did not obtain an adequate fit either (M8-PW: $\Delta\chi2 = 855$; $\Delta df = 150$; $\Delta$RMSEA $= .033$; $\Delta$CFI $= -.068$; $\Delta$TLI $= -.058$). A partial invariance test was executed after observing local misfit. As in the strong invariance tests, the RMSEA met the criteria while the CFI did not (M8p: $\Delta\chi2 = 359$; $\Delta df = 130$; $\Delta$RMSEA $= .003$; $\Delta$CFI $= -.022$; $\Delta$TLI $= -.005$; M8-PWp: $\Delta\chi2 = 348$; $\Delta df = 90$; $\Delta$RMSEA $= .012$; $\Delta$CFI $= -.024$; $\Delta$TLI $= -.018$).

### Invariance tests with method factor

The invariance of the M4 and M4-PW models (that is, models with method factor) was also tested. For the same reasons we pointed out above about the loss of degrees of freedom, the fit indices of these models were almost identical to those obtained in the invariance of M2 and M2-PW.

## Convergent and predictive validity of the scales

Table 3 shows the bivariate correlations between the study scales. SWLS, PA, NA, E and N described among them most of the substantive correlations of this study. Firstly PA, NA and SWLS correlated with a similar magnitude (.5. −.40 and .41). Second, we found the same pattern of Pearson correlations between the adapted short-form scales compared to previous studies with the longer original ones: (a) PA with E (.47) and N (−.40); (b) NA with N (.48) and E (−.19); (c) SWLS with N (−.25), E (.23), and C (.22). C also described a significant but smaller correlation than N and E with NA (−.16). The lowest significant correlation was between PA and O (.13).

We found the same pattern of correlations described above between the Mini-IPIP Spanish PW and the rest of scales: (a) PA with E (.47) and N (−.37); (b) NA with N (.45) and E (−.17); (c) SWLS with N (−.23), E (.25), and C (.24). C also described a significant but smaller correlation than N and E with NA (−.13). And again, the lowest significant correlation was between PA and O (.13). There were no statistical differences between correlations using Fisher's z' transformation at p <.05 (SD $\Delta r_i = 0.03$).

The summary of the factor relations of this study was fully illustrated in the SEM model of Fig. 1 (WLSMV and mentioned CU; $\chi2$ (529) $= 947.65$, RMSEA $= .053$; CFI $= .939$; TLI $= .931$). The variances of PA and NA were explained in a medium to high extent by the personality factors ($R^2_{PA} = .54$, $R^2_{NA} = .62$). And the Life Satisfaction variance was explained medially high mainly by affective factors ($R^2_{LS} = .47$). Again, the same pattern of substantive correlations, fit information and magnitude of variance explained on the

**Table 3  Descriptive, reliability and validity indices.**

|  | SWLS | PA | NA | E | N | A | C | O | E_PW | N_PW | A_PW | C_PW | O_PW |
|---|---|---|---|---|---|---|---|---|---|---|---|---|---|
| PA | .497 | | | | | | | | | | | | |
| NA | −.404 | −.411 | | | | | | | | | | | |
| E | −.231 | .467 | −.192 | | | | | | | | | | |
| N | −.253 | −.404 | .482 | −.216 | | | | | | | | | |
| A | – | – | – | .229 | −.073 | | | | | | | | |
| C | .221 | – | −.163 | – | −.143 | .082 | | | | | | | |
| O | – | .131 | – | .178 | −.074 | .179 | – | | | | | | |
| E_PW | .248 | .472 | −.173 | .943 | – | – | – | .172 | | | | | |
| N_PW | −.229 | −.370 | .448 | – | .876 | – | – | – | – | | | | |
| A_PW | – | – | – | – | – | .924 | – | – | – | – | | | |
| C_PW | .238 | – | −.133 | – | – | – | .925 | – | – | – | – | | |
| O_PW | – | .127 | – | .216 | – | .130 | – | .904 | .197 | – | – | – | |
| i | 5 | 5 | 5 | 4 | 4 | 4 | 4 | 4 | 4 | 4 | 4 | 4 | 4 |
| M | 18.13 | 18.41 | 8.39 | 12.24 | 10.82 | 15.92 | 12.72 | 15.42 | 11.86 | 10.74 | 15.62 | 12.65 | 15.43 |
| SD | 3.65 | 2.78 | 3.11 | 3.56 | 3.20 | 2.69 | 3.54 | 2.97 | 3.28 | 2.82 | 2.73 | 3.22 | 2.85 |
| SK | −0.77 | −0.24 | 1.30 | −0.15 | 0.23 | −0.88 | −0.14 | −0.55 | 0.03 | 0.29 | −0.79 | −0.19 | −0.65 |
| K | 0.89 | 0.85 | 1.63 | −0.33 | −0.24 | 1.25 | −0.62 | 0.11 | −0.10 | −0.09 | 1.61 | −0.44 | 0.38 |
| α | .85 | .83 | .78 | .84 | .65 | .82 | .78 | .79 | .82 | .63 | .86 | .80 | .77 |
| ω | .86 | .84 | .83 | .84 | .67 | .85 | .80 | .81 | .86 | .69 | .95 | .86 | .82 |

**Notes.**
SWLS, The Satisfaction with Life Scale; PA, Positive Affective; NA, Negative Affective; E, Extraversion; A, Agreeableness; C, Conscientiousness; N, Neuroticism; O, Openness; PW, Positively worded version; i, number of items in the scales; $\alpha$, Cronbach's $\alpha$; $\omega$, McDonald's $\omega$.
Pearson's Correlations >.20 or with $p < 0.05$ are shown.

criteria variables was found with the Mini-IPIP PW ($\chi 2$ (529) = 994, RMSEA = .056; CFI = .936; TLI = .928, $R^2_{PA}$ = .53, $R^2_{NA}$ = .56, $R^2_{LS}$ = .47).

Two more reliability indices are shown in the same table; all were good (from .78 to .86) except N (as with composite reliability) which showed a tight but acceptable magnitude of reliability for a short-form.

## DISCUSSION

This study aimed to translate and adapt two versions (balanced and positively worded) of the Mini-IPIP scales to Spanish (*Donellan et al., 2006*), and to examine aspects related to its validity (internal structure, presence of response artefacts, and cross-cultural invariance) by estimating a series of exploratory structural equation models (ESEM) on samples from Chile and the United States.

Our results suggest the following: first, both Mini-IPIP versions (balanced and positively worded) showed congruent properties with theoretical expectations in the form of five clearly identifiable and separable factors. The items demonstrated high convergent validity, as their primary loadings were generally high (>.70) and the cross-loadings were generally low or not significantly different from zero (high discriminant validity). The reliability of the factors was adequate, particularly considering the low number of items per factor.

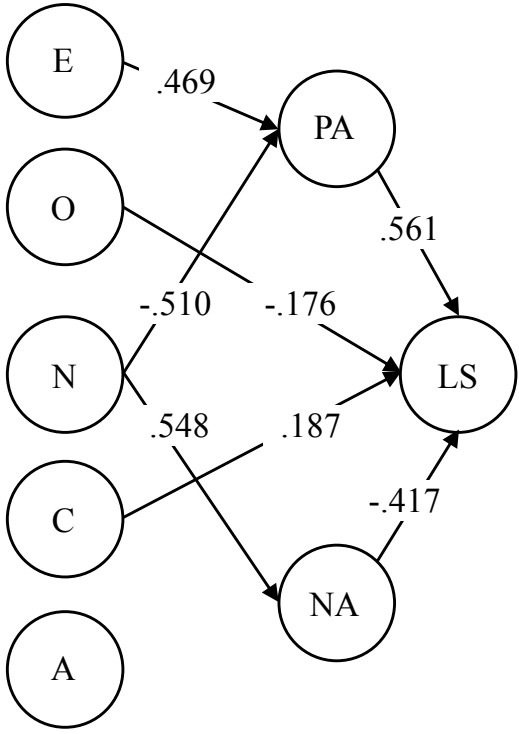

**Figure 1 Estimated SEM model for validity purposes.** Two Mini-IPIP versions, regular and positively worded: validity and cross-cultural invariance E, Extraversion; A, Agreeableness; C, Conscientiousness; N, Neuroticism; O, Openness; LS, The Satisfaction with Life Scale; PA, Positive Affective; NA, Negative Affective; only significant parameters are shown; $R^2PA = .54$, $R^2NA = .62$, $R^2LS = .47$.



Second, after using data screening techniques, modeling a method factor was not relevant. These results are especially interesting because support the use of positively worded items for personality assessment purposes (simpler for the respondent) supplemented by data screening techniques to control the undesired effect of other sources of variability.

Third, both versions (balanced and positively worded) were close to achieving a strong and strict level of invariance with regard to the original version (in English). As in the measurement invariance study performed by *Żemojtel Piotrowska et al. (2017)*, although the CFI difference criterion was not reached, the RMSEA's was. It is necessary to point out that for measurement invariance analysis *Chen (2007)* recommends using the thresholds criteria in a supplemented way (not as added conditions). So, what is the reason for the RMSEA-CFI discrepancy in these invariance tests? It is appropriate to consider that (a) the cut-off values for noninvariance are not "golden rules" and may not be in all generalizable conditions. Most of the fit indices may be sensitive to some conditions such as sample size, factor structure (e.g., one factor, bifactor), factor relations (e.g., orthogonal, oblique), factorial approach (e.g., confirmatory, exploratory), data nature (e.g., continuous, ordinal), estimation methods (e.g., ML, WLSMV) or correlated errors (*Fan & Sivo, 2007*; *Greiff & Scherer, 2018*; *Greiff & Heene, 2017*; *Heene et al., 2012*; *Sass, Schmitt & Marsh, 2014*); (b) RMSEA and CFI assess the adjustment from different perspectives (*Lai & Green, 2016*; *McNeish, An & Hancock, 2018*).
Judging that the invariant tests of this study were conducted from an ESEM approach, with few related factors, ordinal data, correlated errors and WLSMV estimation, we consider acceptable the strong partial and strict partial invariance level for both Mini-IPIP versions. This indicates that language and wording had little effect on the way in which the scale measures the five personality factors, allowing unbiased cross-cultural comparisons (at least between the Spanish and English versions), both in the context of latent variables as well as raw scores. In other words, the differences in the Big Five observed scores between the Spanish-speaking and English-speaking samples were due exclusively to differences in the measured traits, rather than differences in language.

Considering the results after testing the invariance between items with two different wordings and two different languages (i.e., balanced version with English speakers *vs.* positively worded version with Spanish speakers), it is plausible that the positive version of the Mini-IPIP would also reach a better level of invariance if compared to a positively worded version in English.

An exception was the item "seldom feel blue," which demonstrated uniform differential functioning. However, given that there is (apparently) no substantive reason to expect differential functioning for this item, it is difficult to understand the nature of the discrepancy with any certainty without gathering additional data. Nonetheless, at least three hypotheses can be proposed: (a) the differential item functioning obeyed random sample fluctuations that in broader samples would tend to disappear; (b) the item was understood differently by both samples and hence would have to be reformulated; or (c) the DIF was not related to language but rather to other sociodemographic characteristics that were not modeled (for example, differences in the distribution of age or educational level). On the other hand, this item was also problematic in the studies of *Donellan et al. (2006)* and *Baldasaro, Shanahan & Bauer (2013)*. Future research should consider replacing it with another item with more stable properties.

Third, the other short scales in Spanish used in this study, the adapted *PANAS-C10* and the *SWLS*, showed good psychometric properties (reliability and validity) and support the magnitudes of the relationships previously described in the scientific literature among the five personality factors, emotions, and life satisfaction. The relationships between some of the Big Five scales (N and E) and self-reported affective states were medium in magnitude. It should be noted the role of N, which appeared in this and previous studies, related consistently, significantly, and mainly with all criteria variables. Moreover, the relationship of these personality factors with life satisfaction was low. The affective states factors had more relation with life satisfaction than the personality factors. In addition, an ESEM model of two correlated affective factors (positive and negative) was performed. This model showed similar fit indices with or without a method factor ($\Delta\chi2 = -11.95$; $\Delta df = -1$; $\Delta$RMSEA $= -.007$; $\Delta$CFI $= .004$; $\Delta$TLI $= .006$).

Finally, our results supported both Spanish versions (balanced and positively worded). This study provides the first evidence regarding the psychometric properties of a positively worded Big-Five short measure. The PW version showed the best fit indices of the estimated models (Table 1, M2-PW). Because they are simpler to understand and improve the results, we recommend the use of PW items for research purposes if no other justified reason exists.
It is desirable to include items that improve the psychometric properties regardless of their semantic polarity (e.g., content validity), and not only because they are negative. It is also important to point out the possible confusion between trait-variance, acquiescence-variance and social desirability. In the case of the PW version this may be relevant because all items were written in the most socially accepted or natural way. These possible sources of variability may require modeling more than one method factor.

## CONCLUSION

This study has contributed to research on personality measurement by providing the first psychometric properties of a short positively worded inventory in Spanish based on the Big Five personality traits (the Mini-IPIP). One completely positively worded version of the Mini-IPIP and the other balanced (i.e., balance keying) version, showed reliability and validity indices according to theory and prior studies' findings.

Using an exploratory structural factor approach (ESEM), a five-factor structure was modelled after controlling acquiescence by data screening items. In those assessment conditions, (a) the Big Five-dimensional structure was modeled satisfactorily in two different samples (Spanish and English speakers), and (b) modeling a method factor did not improve fit indices. Our results support the recommendation of *Kam & Chan (2018)* about the use of data screening techniques to identify properly careless respondents. Just as in *Greenberger et al. (2003)*, the use of negative items was not psychometrically relevant in our study. In general, the PW version showed slightly better psychometric indices than the balanced one. This result is consistent with the study of *Gnambs & Schroeders (2017)*, where cognitive abilities can explain wording effects (i.e., negatively worded items add an extra difficulty for the participant to correctly understand the content). Therefore, PW short versions of personality used after data screening techniques may be appropriate for future studies.

The demonstration of sufficient measurement invariance with regard to the English version of the Mini-IPIP is particularly interesting for future cross-cultural studies. It is even more interesting to highlight the good psychometric performance shown by the scales regardless of their wording (i.e., the invariance level achieved between the positive version in Spanish vs. the English balanced version). We still have to think about what the causes of the method factors are. In this study, this causes do not appear to be a main cause of misfit.

### Limitations and directions for future research

Some limitations of the study should be considered. Perhaps the most relevant of these is the absence of direct evaluations of criterion validity through the inclusion of the Mini-IPIP in a broader nomological network than the one deduced from the original version. Second, it would be desirable in future studies to contrast the functioning of the Mini-IPIP in Spanish with other broader personality measures in order to verify that its results converge sufficiently. Finally, we would like to point out some considerations made by a reviewer of this manuscript: (a) the Spanish-speaking sample was composed exclusively of university students of a limited age range. The Mini-IPIP-PW may be less appropriate for non-university participants, where individual differences in acquiescent responding would be

more-pronounced (*Rammstedt & Farmer, 2013*). However, the equivalency of the Spanish version with regard to the English-speaking sample with more varied sociodemographic characteristics provides reasonable evidence of the fact that our results are not biased due to the origin of the sample; (b) although the results of the invariance analysis support the equivalence of the studied scales, the process of reducing the English IPIP version to the Mini-IPIP does not have to converge in the same set of items than those that could have resulted from abbreviating the scales with data from participants in Spanish. It would be interesting to carry out studies to analyze if the same set of items converge from different languages; (c) if data control techniques are not taken (*DeSimone, Harms & DeSimone, 2015*; e.g., bogus or instructed items), the use of fully imbalanced scales such as the Mini-IPIP-PW (i.e., only positively worded items) can confound substantive variance with other sources of uncontrolled variance (e.g., the acquiescent response style).

As noted by *Baldasaro, Shanahan & Bauer (2013)*, the strategy to select the items of the Mini-IPIP (those with the largest loading in their theoretical factor) possibly implies that the items represent their respective constructs in quite a narrow way (i.e., lack of content validity). This is supported by the need to relax correlations between residuals of items that are practically equal in wording and content (especially in the Intellect factor, where the two pairs of items are clearly redundant).

In order to continue the validity exercise, we suggest including larger samples and larger variable sets (e.g., academic performance, depression). In future versions of the Mini-IPIP it would be recommended to test new items with greater diversity of content and positive and negative wording, taking advantage of the fact that recent analytical techniques (like ESEM) facilitate the modelling of complex latent structures.

Finally, it is necessary to perform simulations to examine the sensitivity of goodness of fit indexes to the lack of measurement invariance considering other features which are becoming more frequent (e.g., ESEM, WLSMV, ordinal data).

### Funding

This work was supported by the "Fondo Nacional de Desarrollo Científico y Tecnológico" of Chile (FONDECYT No.1151271). The funders had no role in study design, data collection and analysis, decision to publish, or preparation of the manuscript.

### Grant Disclosures

The following grant information was disclosed by the authors:
Fondo Nacional de Desarrollo Científico y Tecnológico: 1151271.

### Competing Interests

The authors declare there are no competing interests.

## Author Contributions

- Agustín Martínez-Molina conceived and designed the experiments, performed the experiments, analyzed the data, contributed reagents/materials/analysis tools, prepared figures and/or tables, authored or reviewed drafts of the paper, approved the final draft.
- Víctor B Arias conceived and designed the experiments, performed the experiments, analyzed the data, contributed reagents/materials/analysis tools, authored or reviewed drafts of the paper, approved the final draft.

## Human Ethics

The following information was supplied relating to ethical approvals (i.e., approving body and any reference numbers):

Ethical approval was obtained from the Bioethics Committee of Universidad de Talca (projects no. 1151271, no. 11140524).

## Data Availability

The raw data are provided in the Supplemental Files.

## Supplemental Information

Supplemental information for this article can be found online at http://dx.doi.org/10.7717/peerj.5542#supplemental-information.

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
