# Peer review of "Balanced and positively worded personality short-forms: Mini-IPIP validity and cross-cultural invariance"

_PeerJ, doi:10.7717/peerj.5542_

## Round 0.1 · original submission · Major Revisions

Thank you for submitting your article to PeerJ. I have now received two reviews and I would like to thank both reviewers for their thoughtful assessments of the manuscript. The reviews are appended below and I won’t reiterate all the comments. However, both reviewers have highlighted important issues that need to be addressed in any revision.

Please ensure you address all reviewer comments; however, I believe that the following comments (that were mentioned by both reviewers) warrant particular attention when revising your manuscript:

1) Both reviewers (and I) found aspects of the procedure section a little unclear and difficult to follow. I would encourage you to clarify this in your revision. Additionally, although it becomes clear in the results section, I would recommend reporting why the English speaking sample was recruited in the methods section, and linking this explicitly to the aims at the end of the introduction.

2) Both reviewers have made suggestions regarding the statistical analyses, and the reporting of these. Please ensure you address these comments clearly in any revision.

3) Both reviewers made fairly substantial comments regarding the positively worded version of the mini IPIP. In general, I found myself agreeing with Reviewer 1’s suggestion of either dropping these analyses or including them in supplementary materials linked to the manuscript (as this might allow a tighter focus on the traditional version of the mini IPIP whilst still keeping the information regarding the positively worded version easily accessible for interested readers). If you do decide to include the positively worded version in the main manuscript (which would also be fine) it will be critical to address Reviewer 2’s comments around the justification of using negatively worded items and the potential biases introduced by an imbalance in positively and negatively keyed items (and the implications of this) and incorporate these analyses more tightly into the background and research aims. I don’t think these analyses warrant an additional paper (particularly given the concerns raised by Reviewer 2).

Thank you again for submitting your article to PeerJ. I hope the reviewers’ comments are helpful in revising your manuscript, and that the points above are useful in focusing your response.

Reviewer 1 ·

Basic reporting

I've made a few small suggestions to improve the writing, but it was generally very strong (please refer to general comments).

Experimental design

The procedure section was unclear but can be easily clarified (see general comments). The design was fairly simple but appropriate given the stated goals.

Validity of the findings

No comment

Additional comments

Overall, I believe this manuscript is likely to make a useful contribution to the study of personality in Spanish proficient / first language samples. Apart from perhaps the first three, my concerns are largely minor and I suspect could be addressed either with small changes to the manuscript or a persuasive case made to the Associate Editor. I list these below.

1. The authors ‘started with’ the Donnellan et al. 20-item measure and translated these items into Spanish to create the mini IPIP. I see one potential issue with this approach. It is quite possible that the results that led Donnellan et al. to retain the items that they retained were driven in part by how the items are written and interpreted in English. In other words, items in English could co-vary for reasons that do not apply when those items are translated to Spanish (or, indeed, responded to by people who are from different countries than the original research). Because the authors did not translate all 50 of the original IPIP items and collect data on all of these, it is impossible to tell from their data whether they have ended up on the best possible short measure.
Having said that, I can see why the authors might be reluctant to revisit the original 50 items; doing so may lead to a slightly different ‘ideal’ set of 20 items, which would then cause difficulty in comparing the new Spanish version of the mini-IPIP to the original English version. Further, given the cross-cultural invariance results, I also would be surprised, should there be a ‘better’ set of 20 translated items, that this better set would be _that_ much better than the 20 they ended on. I therefore see this as perhaps an issue worth raising in the discussion but not a ‘deal breaker’ for this study.
2. The Procedure section of the current manuscript actually does not describe much about the procedure; please provide some more detail. For example, it isn’t clear until you read the results section what the two different samples actually did (I found myself wondering why the US sample was asked to complete a Spanish personality questionnaire, but it became obvious later that they were not).
3. I wasn’t clear on what the contribution of the positively worded version of the mini-IPIP provided. Why would other researchers use that version rather than the traditional version? Both versions appeared to perform well in the latent variable analyses, although I note it’s not easy to compare fit statistics between the two measures since they are not nested models. My suggestion is to consider removing the positively worded version from the manuscript or perhaps present the analysis of it as a supplement or as a short second paper.
4. I wasn’t clear on how the validity criterion variables were chosen. They would seem to be very N and E centric, meaning that they cannot really provide much more than indicative evidence of some of the new mini IPIP scales. If these were simply convenience measures (i.e., they happened to be available in the data set), then I do still think it is worth presenting the results, however, I feel they should be described as such, and suggestions for a larger variable set be made in the discussion so others can continue the validity exercise. Further to this point, please make it clear whether the personality scales were those based on the PW version or the ‘traditional’ version of the mini-IPIP (I apologise if it was there but I missed it).
5. Regarding M2, I was a bit uncomfortable with the description of the decision to allow residuals to correlate being made “a priori” when one of the criteria that were to be fulfilled was that a Modification Index would be large. Unless the criteria for ‘large modification index’ is specified, one could argue that the approach was post hoc rather than a priori.
6. I liked the idea of testing the RI model, but it appeared that the method factor did not do very much to improve fit (what was the estimated factor loading for the method factor, by the way?). Given that, I wondered why the authors retained the method factor when conducting the invariance tests.
7. Please report χ2 and change in χ2 in text for all analyses involving absolute and relative fit.
8. I’d like to suggest an appendix that allows researchers to download the items in a convenient format. I appreciate that such an appendix would double up the content of Appendix A, but Appendix A is a bit unwieldy for researchers wishing to copy and paste the items into their survey, and these days, space restrictions are less of a concern. Further to this, please also provide the precise instructions that the participants received when completing the Spanish mini-IPIP along with the precise response scale that was used.
9. This may have been a problem with my ability to use the manuscript review system, but I was unable to locate Figure 1.
10. Some very minor and specific issues, either typographical or for clarity:
a. I felt the opening sentence was a bit weak; “accounting for a wide range of variables” seemed a bit like a glib ‘throwaway’ line.
b. I suggest replacing “metric” with “psychometric” except with respect to the discussion on invariance testing (the term “metric invariance” is often used to describe a test where factor loadings are fixed).
c. On page 5 paragraph 2, I wasn’t clear what ‘the wording effect’ was. Perhaps a preceding sentence that provides this paragraph with context would assist. E.g., “Many psychometric scales contain a mix of positively and negatively keyed items because…. But….”. I don’t expect you to use my sentence of course, but I just provide it to make my point here more concrete!
d. Please replace “porpoise” with “purpose”.
e. What did “size fraction of sample 1 after attentional control” mean?
f. Under measurement models (page 10), please specify which data were fitted to the big five model.
g. I would argue that the “Cross-cultural invariance” tests/analyses should in fact be referred to as “Cross cultural and language invariance” tests/analyses.
h. “Explained in a medium-high way” seems a bit awkwardly worded. Perhaps “explained to a medium to strong extent”
i. I wasn’t clear on how the last sentence of the first paragraph on page 19 followed from the previous discussion.

·

Basic reporting

The basic reporting in this paper is generally clear, but I have a few suggestions for improving it.

p. 5: “Laverdière, Morin, & St-Hilaire (2013) found poor fit with the confirmatory model, requiring the freeing of correlations between three pairs of residuals because of semantic similarity of items belonging to the same facet.... On short scales, the wording effect is even less known.” It is unclear what the authors mean by “semantic similarity” and “wording effect.” I therefore recommend defining these terms.

p. 5: “The reasons why the creation of negative wording items was usually justified (e.g., avoiding acquiescence) are no longer so clear (Arias & Arias, 2017).” I am not convinced that this statement is supported by the psychometric literature. For example, there is considerable evidence that uncontrolled individual differences in acquiescent responding can bias the observed structure of an item set (e.g., Rammstedt & Farmer, 2013, Psych Assessment; Soto, John, Gosling, & Potter, 2008, JPSP), and can also bias trait-criterion validity associations (Danner, Aichholzer, & Rammstedt, 2015, JRP; Soto & John, in press, Psych Assessment). This is because scales with an imbalance of positively and negatively keyed items confound individual differences in the psychological trait being assessed with individual differences in acquiescent responding. These biases tend to be especially problematic among samples with low levels of formal education, including children and adolescents (Rammstedt & Farmer, 2013; Soto et al., 2008). In the present research, the Mini-IPIP-PW only includes items keyed in the socially desirable direction for each of the Big Five traits. It therefore perfectly confounds substantive trait variance with both acquiescence and social desirability. This strikes me as an important limitation.

To address this issue, I recommend (a) adding a couple paragraphs to the Introduction to summarize objections to scales that include only positively keyed (or only negatively keyed) items, and to respond to these objections, (b) adding a paragraph to the Discussion to acknowledge and discuss the confounding of trait variance with acquiescence variance and social desirability as a limitation of the Mini-IPIP-PW, and (c) expanding the Limitations subsection to acknowledge that the Mini-IPIP-PW may be less appropriate for younger and less well educated samples, due to the possibility of more-pronounced individual differences in acquiescent responding. (I certainly acknowledge that negatively keyed items can have their own limitations and complexities, and the authors are of course welcome to discuss these as well.)

pp. 7, 9, 10: The writing is generally clear, but the authors repeatedly refer to “porpoises” when I believe they mean “purposes.” I recommend editing these sentences.

Experimental design

The research design and analyses are generally well conducted and reported. As the authors acknowledge, the most important limitation is the small number of validity criteria (p. 19). Beyond this, I only have one suggestion for improvement.

Tables 2, 3: The authors report factor loadings, discriminant correlations, and criterion correlations for the Mini-IPIP but not the Mini-IPIP-PW. I recommend adding the Mini-IPIP-PW results to Tables 2 and 3.

Validity of the findings

The interpretation of findings is generally appropriate, but I do have one important suggestion for improvement.

pp. 13, 15, 17: The authors clearly state their criteria for evaluating model fit and measurement invariance (p. 13, lines 315-320). However, they do not apply these criteria consistently when interpreting the present results. Specifically, they conclude that the Mini-IPIP and Mini-IPIP-PW show strong and strict measurement invariance, despite fit statistics below the stated thresholds, as well as non-trivial changes in fit compared with the configural invariance models (pp. 15, 17). To address this inconsistency, I recommend revising either the criteria or the conclusions so that they match with each other.

Additional comments

The Spanish-language Mini-IPIP and Mini-IPIP-PW appear to be reasonably reliable, structurally valid, and brief measures of the Big Five. As noted above, I do have some important recommendations for improving the paper, but on balance I think these measures can be useful tools for personality assessment.

---

## Round 0.2 · Minor Revisions

I would like to again thank both reviewers for their thoughtful consideration of your revised manuscript. You will see that both reviewers have provided some additional comments to be addressed.

Specifically, Reviewer 1 has provided some recommendations to clarify the reporting of the statistical analyses and recommended an additional proofread of the manuscript.

Reviewer 2 has two broader concerns.

First, he suggests that the paper would benefit from a more thorough engagement with the literature on balanced vs. imbalanced item keying, and recommends adding a paragraph about the limitations of positively-worded-only (or negatively-worded-only) scales to the Introduction. Given that PeerJ does not impose word constraints, I believe that incorporating this information should be reasonably straightforward and would strengthen the paper.

Second, Reviewer 2 also noted that the manuscript’s conclusions regarding measurement invariance did not match the stated criteria regarding tests for measurement invariance (changes in fit statistics). I agree with this comment and believe that it needs to be addressed in the revision. In the data analysis section increases in CFI and TLI of less than .01 and decreases in RMSEA of less than .015 were stipulated as indicating invariance; however, changes in these fit statistics clearly exceed these thresholds in the tests of both strong and strict invariance (page 16). The revised manuscript needs to consider this, and given the mismatch with the stated criteria, the conclusions regarding invariance need to justified.

Thank you for submitting your manuscript to PeerJ, and I hope these comments, as well as those of both reviewers, are useful in revising your paper.

Reviewer 1 ·

Basic reporting

Please see general comments

Experimental design

Please see general comments

Validity of the findings

Please see general comments

Additional comments

Overall I am satisfied by the changes to the manuscript. Several issues remain but these are easily fixed:
1. Please report degrees of freedom for the change in chi-square tests
2. Please give the manuscript a proof-read. There were two deleted track-changes (a priori) that remained, and line 367 I believe needed commas added in between the statistics in parentheses). line 458: is probably best not to start a section with "However, it..." Maybe replace with "Several limitations of this study should be acknowledged" or something similar.
3. Figure 1 didn't seem like a CFA model. Rather, it appears to be a SEM as the single headed arrows imply regression of (1) LS on PA, O, C, and NA, (2) NA on N, and (3) PA on E. My understanding of CFA models is that they are models of correlated factors. In any case I don't beleive that SEM is necessary to provide evidence of convergent validity of the new B5 scale.
4. Please add all inter-factor correlations to Table 2 as there is no reason to suppress those and they may be helpful for future analyses (e.g., comparing results of this study to that of future studies using the same measure). I understand that supressing small factor loadings makes FA tables easier to interpret.

·

Basic reporting

The authors have adequately addressed my concerns regarding basic reporting, except for one. Here was my original comment:
* * *
p. 5: “The reasons why the creation of negative wording items was usually justified (e.g., avoiding acquiescence) are no longer so clear (Arias & Arias, 2017).” I am not convinced that this statement is supported by the psychometric literature. For example, there is considerable evidence that uncontrolled individual differences in acquiescent responding can bias the observed structure of an item set (e.g., Rammstedt & Farmer, 2013, Psych Assessment; Soto, John, Gosling, & Potter, 2008, JPSP), and can also bias trait-criterion validity associations (Danner, Aichholzer, & Rammstedt, 2015, JRP; Soto & John, in press, Psych Assessment). This is because scales with an imbalance of positively and negatively keyed items confound individual differences in the psychological trait being assessed with individual differences in acquiescent responding. These biases tend to be especially problematic among samples with low levels of formal education, including children and adolescents (Rammstedt & Farmer, 2013; Soto et al., 2008). In the present research, the Mini-IPIP-PW only includes items keyed in the socially desirable direction for each of the Big Five traits. It therefore perfectly confounds substantive trait variance with both acquiescence and social desirability. This strikes me as an important limitation.

To address this issue, I recommend (a) adding a couple paragraphs to the Introduction to summarize objections to scales that include only positively keyed (or only negatively keyed) items, and to respond to these objections, (b) adding a paragraph to the Discussion to acknowledge and discuss the confounding of trait variance with acquiescence variance and social desirability as a limitation of the Mini-IPIP-PW, and (c) expanding the Limitations subsection to acknowledge that the Mini-IPIP-PW may be less appropriate for younger and less well educated samples, due to the possibility of more-pronounced individual differences in acquiescent responding. (I certainly acknowledge that negatively keyed items can have their own limitations and complexities, and the authors are of course welcome to discuss these as well.)
* * *
In response, the revised manuscript includes a brief discussion of how a mix of positively and negatively worded items can introduce the need for a method factor (p. 5), as well as a brief acknowledgement that the Mini-IPIP-PW may be less appropriate for non-university participants (p. 21). However, I continue to believe that the paper would benefit from a more thorough discussion of balanced vs. imbalanced item keying. In particular, I suggest adding a paragraph about the limitations of positively-worded-only (or negatively-worded-only) scales to the Introduction (p. 5): Fully imbalanced scales perfectly confound substantive personality variance with acquiescent response style. It is therefore impossible to determine whether a high score (or low score) on the scale is due to the respondent’s (a) true standing on the trait being measured, or (b) tendency to consistently agree (or disagree) with items regardless of their content (i.e., acquiescence). This is a long-standing issue in the psychometric literature that is relevant to the Mini-IPIP and Mini-IPIP-PW, and it deserves additional consideration in this manuscript. Readers choosing between the Mini-IPIP and Mini-IPIP-PW should be made aware of the psychometric advantages and disadvantages of each version.

Experimental design

The authors have adequately addressed all of my concerns regarding experimental design.

Validity of the findings

In my original review, I noted that the manuscript’s conclusions regarding measurement invariance did not match the stated criteria regarding tests for measurement invariance. That continues to be the case in the revised manuscript. Specifically, the authors conclude that the Mini-IPIP obtained strong invariance but not strict invariance (p. 16). However, the fit statistics for strong invariance and strict invariance are quite similar to each other, and neither meets the stated criteria for testing invariance (p. 14). In particular, both the Mini-IPIP and Mini-IPIP-PW show substantial decreases in CFI (delta > .035) and TLI (delta > .022) for both strong and strict invariance.

To address this issue, I recommend further revisions to bring the conclusions regarding measurement invariance into alignment with the stated criteria.

Additional comments

This revised manuscript has thoughtfully addressed some, but not all, of my concerns with the original manuscript. I therefore recommend further revisions before accepting the paper for publication.

---

## Round 0.3 · accepted · Accept

Thank you for your revised submission and your responses to the reviewers' comments. I am pleased to accept your manuscript for publication in PeerJ. However, I do have some very minor suggestions that I would recommend you make for readability, which can be addressed while in production.

1) Without knowing the details in the manuscript, the title is a bit confusing. I would recommend rewording it along the lines of " Validity and cross-cultural invariance of balanced and positively worded versions of the Spanish Mini-IPIP" or something similar.

2) Related to this, throughout the document you alternate between "balance" and "balanced" wording. I recommend being consistent in your wording - and suggest "balanced wording" would read more clearly.

3) While the instructional manipulation check is important, I don't think it needs to be mentioned in the abstract - as at this point a reader won't know what this actually refers to. Also, you need to specifcally limit your conclusions to the positively worded version of the IPIP

4) I think you are overstating the findings in lines 492 - 496 (page 20). Specifically, I think it is best to avoid language like "remarkable achievement" and statements like "it is certain that the positive version of the mini-IPIP would reach a better level of invariance ..." While this is likely the case, you can't claim that this is certain.

5) I would recommend removing the "Prospective" subheading on page 24 and instead change the "Limitations" subheading (page 23) to "Limitations and Directions for Future Research".

6) For point c on page 24, about confounding substantive and other sources of variance, I would recommend including a reference.

7) Finally, this version of the paper is improved after the proof-reading recommended previously by Reviewer 1. However, I would again recommend a final very close reading, specifically focusing on consistency of language use throughout the document (e.g. balance v balanced)